# SMILES2VEC: PREDICTING CHEMICAL PROPERTIES FROM TEXT REPRESENTATIONS

**Garrett B. Goh [1,*], Nathan Hodas [2], Charles Siegel [2], Abhinav Vishnu [1],**

[1] Advanced Computing, Mathematics & Data Division, Pacific Northwest National Lab
[2] Computing & Analytics Division, Pacific Northwest National Lab
garrett.goh@pnnl.gov

## ABSTRACT

Chemical databases store information in text representations, and the SMILES format is a universal standard used in many cheminformatics software. Encoded in each SMILES string is structural information that can be used to predict complex chemical properties. In this work, we develop SMILES2vec, a deep RNN that automatically learns features from SMILES strings to predict a broad range of chemical properties, including toxicity, activity, solubility and solvation energy. Furthermore, we trained an interpretability mask for SMILES2vec solubility prediction, which identifies specific parts of a chemical that is consistent with ground-truth knowledge with an accuracy of 88%, demonstrating that neural networks can learn technically accurate chemical concepts.

## 1 INTRODUCTION

In chemistry, contemporary deep learning (DL) models developed to predict chemical properties, are based almost exclusively on engineered chemical features. Goh et al. (2017a) While such an approach utilizes existing knowledge, it also limits the search space of potentially developable representations, which can be an issue if feature engineering is suboptimal. To date, there has been limited work on representation learning driven DL models in chemistry. Some exploration has been done in using molecular graphs, Duvenaud et al. (2015); Kearnes et al. (2016) and image data. Goh et al. (2017c;d). Chemical information can also be encoded in a text format (SMILES), and we acknowledge there has been some prior work in this direction. Jastrzebski et al. (2016); Bjerrum (2017).

Our contributions to the field of DL and chemistry is as follows: *(i) we develop SMILES2vec, a common RNN architecture for interpreting the SMILES "chemical language" and evaluated its performance on a much broader set of chemical properties, in the process demonstrating that SMILES2vec outperforms contemporary MLP models that uses engineered features, and (ii) using an interpretability mask, we explain how SMILES2vec makes solubility prediction, and obtained good correlation to ground-truth chemical knowledge.*

## 2 DATA

SMILES2vec is designed as a general-purpose chemical property predictor, and was tested on a broad range of properties, Wu et al. (2017), including the Tox21, HIV and FreeSolv dataset (Table 1). We also used the ESOL solubility dataset as a proof-of-concept for interpreting SMILES2vec. The data was prepared by mapping unique characters in the SMILES string to one-hot vectors. Zero padding was applied both the left and right of the string to construct uniform entries that were 250 characters long. The data splitting is similar to that reported in previous work, and random 5-fold cross validation was used for training. Goh et al. (2017c)

| Dataset | Property | Task | Size |
|---|---|---|---|
| Tox21 | Non-Physical (Toxicity) | Multi-task binary classification | 8014 |
| HIV | Non-Physical (Activity) | Single-task binary classification | 41,193 |
| FreeSolv | Physical (Solvation) | Single-task regression | 643 |
| ESOL | Physical (Solubility) | Single-task regression | 1128 |

Table 1: Characteristics of the 4 datasets used to evaluate the performance of SMILES2vec.

## 3 EXPERIMENTS

### 3.1 NEURAL NETWORK TRAINING & DESIGN

SMILES is a "chemical language" Weininger (1988) that encodes structural information into a compact text representation. In this work, we develop SMILES2vec, a RNN model (Figure 1) for a sequence-to-vector "translation", where the sequence is a SMILES string and the vector is the property to be predicted. SMILES2vec was trained using Tensorflow Abadi et al. (2016) with GPU acceleration using NVIDIA CuDNN libraries. Chetlur et al. (2014) The network was created and executed using the Keras 2.0 functional API interface Chollet et al. (2015). We used the RMSprop algorithm Hinton et al. (2012) to train for 250 epochs using the standard settings with early stopping.

As SMILES is a fundamentally different language, commonly-used techniques (e.g. embeddings like Word2vec Mikolov et al. (2013)) cannot be directly adapted for use in the chemical sciences. Therefore, a substantial component of our work is in the design of a SMILES-specific RNN model. We explore the RNN model's architecture class, which includes high-level design choices, such as the type of units used, type of layers, arrangement of layers, etc. LSTMs and GRUs are the two major RNN units used in the literature, and form the basis of two architectural classes as illustrated in Figure 1. In addition, we explored the utility of adding a 1D convolutional layer between the embedding and GRU/LSTM layers. We used a Bayesian optimizer, SigOpt Dewancker et al. (2016) to optimize the hyperparameters related to the neural network topology.

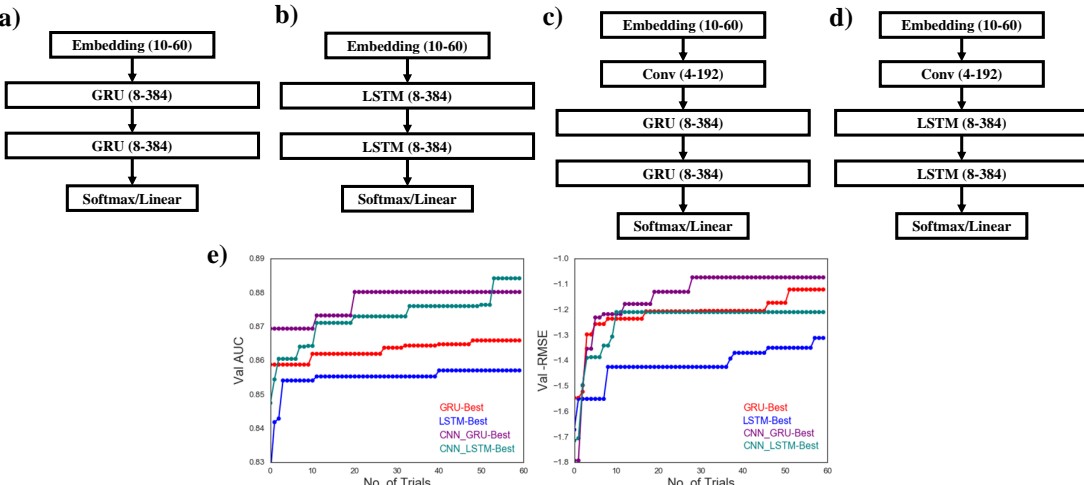

Figure 1: Illustration of the 4 architectural classes investigated, (a) GRU, (b) LSTM, (c) CNN-GRU and (d) CNN-LSTM. Number of units explored is indicated in parenthesis. (e) Results of Bayesian optimization of the hyperparameters of the 4 architectural classes for the Tox21 nr-ahr classification and Freesolv regression tasks

The results of the Bayesian optimization across all 4 classes are as indicated in Figure 1. For Tox21 nr-ahr classification, we observed that an additional convolutional layer between the embedding and GRU/LSTM layers improved model performance relative to their counterparts, and the best performing model was the CNN-LSTM class, with CNN-GRU trailing slightly behind. For FreeSolv regression, we observed that GRU-based networks outperform LSTM-based networks. Taking into considerations for generalization to other type of chemical properties, we selected the CNN-GRU architectural class with the best hyperparameters for the remainder of this work.

## 3.2 GENERALIZATION OF SMILES2VEC MODELS

In the development of SMILES2vec, only a fraction of the 4 datasets identified for this work was used to optimize the hyperparameters; the HIV dataset was not included, and 11 out of 12 toxicity tasks were not included. Hence, this section also determines how generalizable the Bayesian optimized network design will be to other chemical tasks.

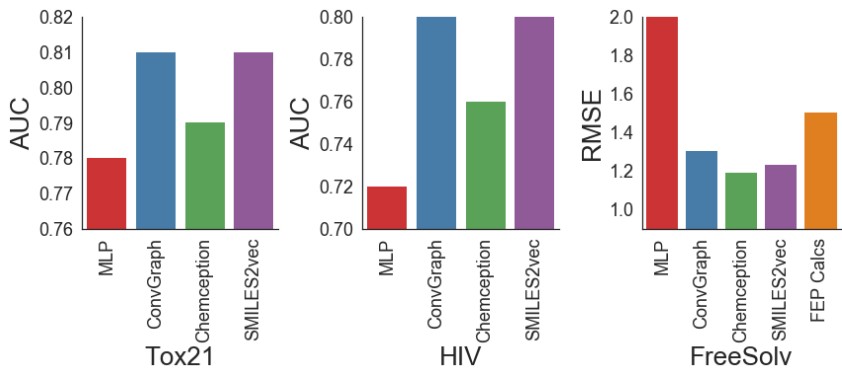

Figure 2: Performance of SMILES2vec against contemporary DL models trained on engineered features, image and graph data. For Tox21 and HIV, higher AUC is better. For FreeSolv, lower RMSE is better.

Applying a recently developed pre-training approach, Goh et al. (2017b) on the final SMILES2vec model, the following validation performance metrics were obtained: AUC of 0.81 for the Tox21 dataset, AUC of 0.80 for the HIV dataset, and RMSE 1.2 kcal/mol for the FreeSolv dataset. Next, we compared the performance of SMILES2vec against contemporary DL models, including a baseline MLP model that uses engineered features, Wu et al. (2017) a chemistry-specific molecular graph convolutional neural network, Wu et al. (2017) and Chemception, a deep CNN that uses images. Goh et al. (2017d). The results are presented in Figure 2. We observed that baseline MLP models performed the worst. SMILES2vec outperformed Chemception in classification tasks, but slightly underperformed in regression tasks. In addition, SMILES2vec also outperformed first-principles models for computing solvation free energy. Against convolutional graphs, which is the current state-of-the-art for many chemical tasks, SMILES2vec either matches for classification tasks or outperforms for regression task.

Lastly, we trained an interpretability mask for SMILES2vec to identify specific parts of the SMILES text that is responsible for the network's prediction. We used the ESOL solubility dataset, where ground-truth knowledge exist in the chemistry literature. Specifically, chemicals are soluble because of the presence of O and N atoms (characters), and are insoluble if there is too many C atoms. With this ground truth labeling in expected atoms, we evaluated that the top-3 accuracy of SMILES2vec interpretability is 88%, thus demonstrating that neural networks can learn technically accurate chemical concepts.

## 4 CONCLUSION

In this paper, we develop SMILES2vec, the first general-purpose deep neural network that uses chemical text data (SMILES) for predicting chemical property, with an explanation mask that improves interpretability. By performing extensive Bayesian optimization experiments, we identified a specific CNN-GRU neural network architecture that is effective in predicting a wide range of properties. SMILES2vec achieved a validation AUC of 0.81 and 0.80 for Tox21 toxicity and HIV activity prediction respectively, and a validation RMSE of 1.2 kcal/mol and 0.63 for solvation energy and solubility. SMILES2vec outperforms baseline DL MLP models and is competitive against state-of-the-art convolution graph models. In addition, using an interpretability mask, we calculated that SMILES2vec's solubility prediction localizes on the correct atoms(characters) with a top-3 accuracy of 88%. Identification of such atoms and their relationship to solubility is a key first-principles concept, which SMILES2vec was able to discover on its own, suggesting its value as an interpretable tool for DL-driven chemical design.

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
