# OpenReview forum: "SMILES2vec: Predicting Chemical Properties from Text Representations"
_ICLR.cc/2018/Workshop — Reject_

### Official Review · AnonReviewer3 · 2018-02-26
**Why use SMILES when one can use graph convolutions?**

**Rating:** 5
**Confidence:** 5

**Review:**

The SMILES format was developed as a quasi-human-readable string representation for the graph structure of small molecules. While it has proved useful for the efficient communication in chemistry, it is unclear what advantage there is in developing machine learning systems that operate on this representation directly, rather than on the graph structure it encodes. Constraining the learning problem via the neural network architecture can be important to acheiving good performance with deep learning systems; by requiring the network to learn the relationship between the SMILES encoding and the graph structure, and without offering any new information, I would expect this approach to perform worse than the graph convolution approach in general. The authors show similar performance using graph convolution and SMILES-based approaches, but I am skeptical that it would hold up under further experiments or additional hyperparameter tuning. In addition, wouldn't using a graph convolution architecture result in better interpetability?

Other work has explored the use of recurrent neural networks that take SMILES as input. The main contribution of this work is to apply the idea to a different set of chemoinformatics tasks. The results are interesting but incremental.

Liu, Bowen, et al. "Retrosynthetic reaction prediction using neural sequence-to-sequence models." ACS central science 3.10 (2017): 1103-1113.

Fooshee, David, et al. "Deep learning for chemical reaction prediction." Molecular Systems Design & Engineering (2018).

---

### Official Review · AnonReviewer1 · 2018-02-28
**Neither novelty nor importance made clear**

**Rating:** 4
**Confidence:** 2

**Review:**

This paper presents and compares simple embedding models for chemical structures from a textual description of those structures (SMILES).  The simple models seem to perform better than more elaborate alternatives, and comparably to state of the art.

As someone who is not an expert in this field (computational chemistry) it is hard for me to assess the novelty of this work. The authors compare to several recent approaches, some of which are their own.  Moreover, the scores that they present seem to rely on a combination of the present method and their previous method, but this combination is not clearly explained ("Applying a recently developed pre-training approach, Goh et al. (2017b) on the final SMILES2vec model, the following validation performance metrics were obtained: AUC of 0.81 for the Tox21 dataset, AUC of 0.80 for the HIV dataset, and RMSE 1.2 kcal/mol for the FreeSolv dataset.")

They compare against Chemception, their previous deep CNN that uses images, but again the takeaway from this comparison is unclear since the methods patently use different inputs.

Pros
- simple method seems to obviate need for domain-specific knowledge
- seems to perform well on three prediction / regression tasks

Cons
- authors do not explain the point of the task to lay readers (why are chemical structures encoded in text?)
- authors do not explain the nature of the inputs / outputs to lay readers
- authors do not explain very convincingly the wider implications of the results to lay readers
- novelty unclear given that the authors themselves did much of the prior comparison work
- the ICLR workshop is looking for "late-breaking developments, very novel ideas and position papers". I am not convinced (at this stage) that this abstract falls into any of those categories.

---

### Official Review · AnonReviewer2 · 2018-03-12
**Potentially promising work but the presentation is not clear**

**Rating:** 5
**Confidence:** 3

**Review:**

The presented work is potentially interesting but I am reluctant to strongly support it. It is not clear to me what the proposed work is actually doing. The authors present a DNN that takes as input the chemical text data (in the SMILES format) and predict chemical property. For clarity, it is important that the authors expand what this means, i.e. provide an example of SMILE syntax and predicted property.

In terms of originality, I am not an expert in DL for chemistry. It seems to me though that without presenting the actual task in a clear way it is hard to have a saying on that.

The work has a potential, but as I said, the author need to present it in a more thorough manner.

---

### Decision · Program_Chairs · 2018-03-20
**ICLR 2018 Workshop Acceptance Decision**

**Decision:**

Reject

**Comment:**

Based on the reviews, this paper has not been accepted for presentation at the ICLR workshop. However, the conversation and updates can continue to appear here on OpenReview.